# JELLY BEAN WORLD:
# A TESTBED FOR NEVER-ENDING LEARNING

**Emmanouil Antonios Platanios**[*]**, Abulhair Saparov**[*] **& Tom Mitchell**
Machine Learning Department, Carnegie Mellon University, Pittsburgh, PA 15213, USA
{e.a.platanios,asaparov,tom.mitchell}@cs.cmu.edu

## ABSTRACT

Machine learning has shown growing success in recent years. However, current machine learning systems are highly specialized, trained for particular problems or domains, and typically on a single narrow dataset. Human learning, on the other hand, is highly general and adaptable. *Never-ending learning* is a machine learning paradigm that aims to bridge this gap, with the goal of encouraging researchers to design machine learning systems that can learn to perform a wider variety of inter-related tasks in more complex environments. To date, there is no environment or testbed to facilitate the development and evaluation of never-ending learning systems. To this end, we propose the *Jelly Bean World* testbed. The Jelly Bean World allows experimentation over two-dimensional grid worlds which are filled with items and in which agents can navigate. This testbed provides environments that are sufficiently complex and where more generally intelligent algorithms ought to perform better than current state-of-the-art reinforcement learning approaches. It does so by producing non-stationary environments and facilitating experimentation with multi-task, multi-agent, multi-modal, and curriculum learning settings. We hope that the Jelly Bean World will prompt new interest in the development of never-ending learning, and more broadly general intelligence.

## 1 INTRODUCTION

Machine learning has witnessed growing success across a multitude of applications over the past years. However, despite these successes, current machine learning systems are each highly specialized to solve one or a small handful of problems. They have much narrower learning capabilities as compared to humans, often learning just a single function or model based on statistical analysis of a single dataset. One reason for this is that current machine learning paradigms are restricted and specialized to a particular problem and/or dataset. An alternative learning paradigm that more closely resembles the generality, diversity, competence, and cumulative nature of human learning is *never-ending learning* (Mitchell et al., 2018). The thesis of never-ending learning is that *we will never truly understand machine learning until we can build computer programs that, like people: (i) learn many different types of knowledge or functions, (ii) from years of diverse, mostly self-supervised experience, (iii) in a staged curricular fashion, where previously learned knowledge enables learning further types of knowledge, and (iv) where self-reflection and the ability to formulate new representations and new learning tasks enable the learner to avoid stagnation and performance plateaus.* Building computer programs with these properties necessitates well-defined and robust ways to evaluate whether a system is indeed capable of never-ending learning. However, there are currently no ways to achieve that. There only exists one large-scale case study on never-ending learning with the Never-Ending Language Learning (NELL) system by Mitchell et al. (2018), which uses the internet as the environment with which the system interacts. While the internet does have significant complexity, it is unwieldy to use as a testbed. It is very difficult to focus on a particular aspect of the system or the environment, or to tweak the algorithm and restart experiments to observe the effects of changes. Furthermore, oftentimes tasks require manual annotation which can be very expensive. Thus, a good testbed for never-ending learning (and machine learning more generally) needs to provide the experimenter with a high degree of control. To this end, we propose a novel evaluation framework—the *Jelly Bean World (JBW)*—that can enable and facilitate research towards the goal never-ending learning. We have designed the JBW to be highly versatile, enabling evaluation of systems that have any number of the aforementioned abilities.

We consider never-ending learning in the context of reinforcement learning. Let $s_t \in \mathcal{S}$ denote the state of the environment at time $t$, $a_t \in \mathcal{A}$ denote the action performed by the learning agent at

---

[*]Equal contribution (listed in alphabetical order).

time $t$, $\omega_t \in \Omega$ denote the observation of the world that the learning agent receives at time $t$, and $r_t \in \mathbb{R}$ denote the reward provided to the learning agent at time $t$. The distribution of the next state of the world $s_t \sim T(s_{t-1}, a_{t-1})$ has the Markov property (i.e., it depends only on the previous state and action) and the initial state of the world is given by a distribution $s_0 \sim W$. The observation $\omega_t \sim O(s_t)$ depends only on the current state of the world (perhaps deterministically). The reward $r_t$ is given by a function $R(s_{t-1}, a_{t-1}, s_t)$ of the current state, the previous state, and the previous action taken. The environment is a tuple containing all these elements $\mathcal{E} \triangleq (W, T, O, R)$. Then, the goal of reinforcement learning is to find a learning algorithm $\pi$ that, given the history of previous observations, actions, and rewards, outputs the next action so that the obtained reward is maximized. We deliberately blur the distinction between the policy and the algorithm that learns the policy, which is why we call $\pi$ a "learning algorithm."

This formalism does not distinguish between learning algorithms that are highly specialized to a single task and learning algorithms that are capable of learning a wide variety of tasks and adapting to richer and more complex environments, which are hallmarks of *general intelligence*. In order to more formally describe general intelligence, we posit that there is an underlying measure of *complexity* of the environment $\mathcal{E}$ such that: (i) highly specialized and non-general learning algorithms can perform well in environments with low complexity, but (ii) environments with high complexity require successful learning agents to possess more general learning capabilities. It is in these more complex environments where we can characterize never-ending learning. We can formalize this notion of complexity by letting $\pi^*$ be the (computable) learning algorithm that maximizes expected reward in an environment $\mathcal{E}$. Then we define the complexity of $\mathcal{E}$ to be the length of the shortest program (Turing machine) that implements $\pi^*$:

$$\text{complexity}(\mathcal{E}) = \min\{|T| : T \text{ is a Turing machine that implements } \pi^*\}$$

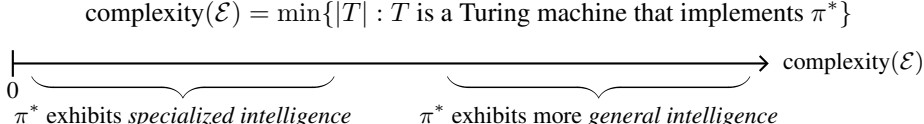

We can equivalently define $\text{complexity}(\mathcal{E}) = K(\pi^*)$, where $K(\cdot)$ is the *Kolmogorov complexity* and is related to the *minimum description length* and *minimum message length* (Nannen, 2010; Kolmogorov, 1963). The Kolmogorov complexity of the environment $K(\mathcal{E})$ is bounded below by $K(\pi^*)$ minus a constant. This bound is shown in Section A.1 of the appendix.

Never-ending learning is in many respects similar to *lifelong learning*, also called *continual learning* (Chen & Liu, 2018). Like never-ending learning, lifelong learning is distinguished from multi-task learning by the never-ending nature of the learning problem. However, in never-ending learning, and unlike multi-task learning and lifelong learning (to the best of our understanding), a well-defined set of tasks is not assumed a priori. Rather, never-ending learning is more similar to real-world settings in this respect, where the notion of a task or subtask naturally emerges from the complexity of the environment, and the distinction between tasks is not always so sharp. Regardless, due to their similarities, a good testbed for never-ending learning will also be a good testbed for lifelong learning.

In contrast to most popular reinforcement learning settings, never-ending learning focuses on environments with *high complexity*. In never-ending learning, agents can only exist in a single reset-free environment (i.e., we explicitly disallow the agent $\pi$ from learning across multiple episodes or in multiple environments, which is closer to human learning). We require $\pi$ to only have access to a single episode. During its lifetime, $\pi$ can only use the information provided by its past observations $\{\omega_t\}$ and actions $\{a_t\}$ in a single world. Thus, never-ending learning explicitly removes the distinction between training and testing that is common to many other classical machine learning paradigms. Additionally, note that in the general reinforcement learning formalism, $s_t$ can contain information about $t$, and the reward function $R$ can be time-varying, thus rendering the environment non-stationary. In never-ending learning, we are interested in the full generality of non-stationary environments, as the assumption of stationarity is not realistic in even simple adversarial and multi-agent settings. We thus argue that a testbed for never-ending learning should have the following properties:

1. **Non-Episodic:** It should disallow agents from resetting the environment and "retrying". The testbed should also force them to only learn within a single environment (i.e., not transfer information across environments). This is in contrast with most popular reinforcement learning environments and, as we show in Section 4, poses significant challenges to existing algorithms.

2. **Non-Stationary:** The testbed should allow for easy experimentation with non-stationary environments, where the reward $R$ can depend on time.

**Table 1:** Existing reinforcement learning environments positioned based on our desired properties.

| Environment | Non-Episodic | Non-Stationary | Multi-Task | Multi-Modal | Controllable | Efficient |
|---|---|---|---|---|---|---|
| Atari and Retro Games (Bellemare et al., 2013), (Pfau et al., 2018) | ✗ Games end when the player wins or loses | ✗ The game mechanics are stationary | ✗ Each game has a single fixed reward function | ✗ Agents only observe the game video frames | ✗ Modifying the task complexity/richness is not possible | ~ Can run on small machines but models are slow to train |
| Continuous Control (Duan et al., 2016), (Todorov et al., 2012) | ✓ Some tasks are non-episodic (e.g., swimmer) | ✗ Stationary rewards and environments (i.e., physics) | ✓ Some of the tasks have interesting hierarchical structures | ✗ Agents only observe positional information and joint angles | ✗ The tasks and environments are non-configurable | ~ Efficiency varies widely across tasks |
| Evolutionary Robotics (Mouret & Doncieux, 2012) | ✗ Episodic in a finite world | ✗ Stationary environments (i.e., physics) | ✗ Only navigation goals are supported | ✓ Multiple different kinds of sensors | ✓ Configurable using XML | ✓ Fast 2D simulation written in C++ |
| BabyAI (Chevalier-Boisvert et al., 2018) | ✗ Episodic in a finite world | ✗ Fixed set of levels and rewards | ✓ Handful of tasks to perform | ✓ Agents given visual input and instructions | ✗ Existing levels are not configurable | ✓ Built on fast MiniGrid simulator |
| Adversarial Games like Go (Silver et al., 2017), StarCraft (Vinyals et al., 2019), and Dota (OpenAI, 2019) | ✗ Games end when the player wins or loses | ✓ Non-stationary (without assumptions about the adversaries) | ✗ Each game has a single fixed reward function | ~ Agents observe the game video frames and the game state | ~ There is limited control over things like the adversary's competence | ✗ Experiments are typically extremely computationally expensive |
| DeepMind Lab (Beattie et al., 2016) | ✗ Levels have a time limit | ~ Levels have different rewards but same physics | ~ Levels have predefined rewards | ✗ Agents only observe the game video frames | ✗ The complexity of each level is fixed | ~ Requires rendering of a 3D world |
| Malmö and MineRL (Johnson et al., 2016), (Guss et al., 2019) | ~ Tasks have a pre-specified time limit but that is typically very long | ✗ Stationary rewards and map generation is based on Perlin noise | ✓ Supports 6 complex tasks but also allows for new ones | ✓ Agents observe the game video frames and the game state | ✗ Modifying the task complexity/richness is difficult and expensive | ✗ Requires rendering of a 3D world and slow training of large models |
| **Jelly Bean World** (Proposed Environment) | ✓ Agents live "forever" in an infinite open world | ✓ The rewards and the world can both be non-stationary | ✓ Composable and dynamic tasks are supported | ✓ Vision and scent are designed to be complementary | ✓ Modifying the task complexity/richness is very easy | ✓ Experiments can run efficiently on small machines |

3. **Multi-Task:** It should support settings in which reward is maximized not by learning how to perform a single task repetitively, but by learning how to perform a general variety of tasks, and learning how to switch between them and/or combine them to better perform other tasks (e.g., by composing them). We posit that, at a sufficiently high level of task complexity, optimal learning agents will be required—either explicitly or implicitly—to perform abstract reasoning over concepts and make informed decisions about actions in the environment.

4. **Multi-Modal:** It should support multiple data modalities that agents receive as input. These modalities should not contain the same information, but rather be complementary to each other so that the agents are forced to learn from diverse types of experiences. Multi-modality provides yet another way to increase the complexity of the world.

5. **Controllable:** It should be easy for experimenters to modify the complexity and richness of the learning problems in the testbed, make changes to it, and restart it (e.g., as opposed to NELL).

6. **Efficient:** It should run on readily available hardware and allow for quick experimentation. Ideally, we should not have to wait for days, weeks, or months (e.g., NELL) to obtain results.

7. **Reproducible:** It should make it easy to reproduce results and experiments, which would facilitate scientific research. This also requires that it allows for seamlessly saving and loading state and for reproducing results outside the environment in which they were first obtained. The testbed should also not require access to specialized hardware, which can be expensive.

Many of the above properties are means to increase the complexity of the world. These properties are in fact very closely related to the characteristics of "AGI Environments, Tasks, and Agents" outlined by Laird & Wray III (2010) and later refined by Adams et al. (2012). The proposed JBW has all of these properties. It aims to provide an easy way to create sufficiently *complex* environments allowing researchers to experiment with never-ending learning, while remaining simple enough to control the problem and enable rapid prototyping. The JBW is a two-dimensional grid world with simple physics, but is extensible enough to admit a wide variety of complex and inter-related tasks. We present a

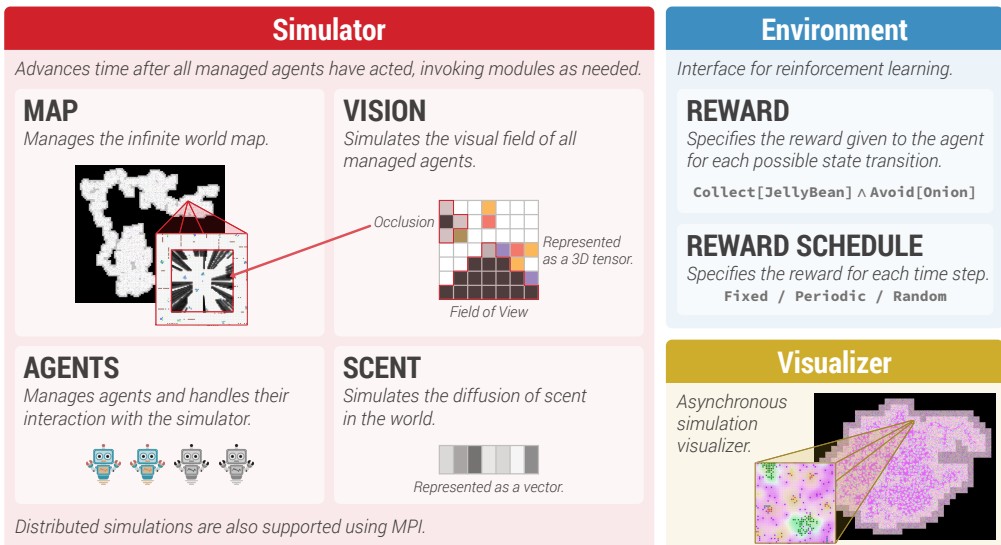

**Figure 1:** Overview of the modules comprising the Jelly Bean World.

comparison with related work in Table 1, showcasing the ways in which the JBW is a novel and highly versatile evaluation framework. The JBW is written in C++ and we provide C, Python, and Swift APIs, and is available at `https://github.com/eaplatanios/jelly-bean-world`. It is also worth noting that the JBW has already been used for the instruction of the "Deep Reinforcement Learning" and "Never-Ending Learning" graduate courses at Carnegie Mellon University.

## 2 DESIGN

The Jelly Bean World (JBW) consists of the following main modules (illustrated in Figure 1): (i) the *simulator*, which comprises the central component (the other modules only interact with the simulator), (ii) the *environment*, which provides a simple interface for performing reinforcement learning experiments in the never-ending learning setting as well as utilities for evaluating never-ending learning systems, and (iii) the *visualizer*, which provides the ability to visualize and debug the behavior of learning agents. Note that the visualizer is completely asynchronous and can be attached, reattached, and detached to and from existing simulator instances, without affecting the simulations.

### 2.1 SIMULATOR

The simulator manages a *map* and a set of *agents*. At a high-level, the map is an *infinite* two-dimensional grid where each grid cell can contain items (e.g., jelly beans and onions) and/or agents. Each item has a *color* and a *scent* that agents can perceive. Each agent has a direction and a position, and can navigate the world map and collect or drop items. The action space of each agent is: to turn, move, collect items, drop items, or do nothing. The action space is configurable and can be constrained by the user. These constraints are described later in this section. Time in the simulator is discrete, and all agent-map interactions are *turn-based*, meaning that the simulator will first wait for all managed agents to request an action and will then simultaneously execute all actions and advance the current time. Thus, the simulator also controls the passage of time.

**Map.** In order to truly support never-ending learning, we have designed the JBW map to be *infinite*, meaning that it has no boundaries and agents can keep exploring it forever. To achieve this, the map is a procedurally-generated two-dimensional grid. We simulate it by dividing it into a collection of disjoint ($P \times P$)-sized patches and only generating patches when an agent moves sufficiently close to them. The map also contains items of various types which are distributed according to a pairwise-interaction point process over the two-dimensional grid (Baddeley & Turner, 2000). More specifically, for a collection of items $\boldsymbol{I} \triangleq \{I_0, \dots, I_m\}$, where $I_i = (x_i, t_i)$, $x_i \in \mathbb{Z}^2$ is the position of the $i^{th}$ item, $t_i \in \mathcal{T}$ is its type, and $\mathcal{T}$ is the set of all item types:

$$p(\boldsymbol{I}) \propto \exp\left\{\sum_{i=0}^{m} f(I_i) + \sum_{j=0}^{m} g(I_i, I_j)\right\}, \tag{1}$$

where $f(I_i)$ is the *intensity* of item $I_i$ and $g(I_i, I_j)$ is the *interaction* between $I_i$ and $I_j$, which are provided as part of the item's type. The intensity function characterizes the (log) probability of the

existence of an item $I_i$ independent of other items in the world. The interaction function can be understood as a description of the (log) probability of the existence of an item $I_i$ given the existence of all other items $I_j$. For example, the interaction function can be used to increase the log probability of an item when it appears near other items, producing a clustering effect. Since the world is subdivided into $(P \times P)$-sized patches, the maximum distance of interaction between items is $P$.

**Item Types.** Each item type $t \in \mathcal{T}$ defines the following:

– Color: Fixed-size vector specifying the item color.
– Scent: Fixed-size vector specifying the item scent.
– Occlusion: Occlusion of an item (relevant to the vision modality, described later in this section).
– Intensity Function: Maps from item locations to real values.
– Interaction Functions: Collection of functions that map from pairs of item locations to real values. The collection contains one function for each item type.

The number of item types and their properties are configurable. The specific parametric forms for the intensity and interaction functions that are supported are described in Section A.2 of the appendix. Note that each item type also specifies additional properties that are described later in this section.

**Procedural Generation.** When the simulator is instantiated the map is empty (i.e., no patches have been generated). Whenever a new agent is added to the simulator, a patch centered at its location is generated. In addition, whenever an existing agent moves sufficiently close to a region where no patch exists, a new patch is generated. The patch generation process consists of two main steps: (i) add a new empty $(P \times P)$-sized patch to the collection of map patches (note that the new patch will be neighboring at least one existing patch and that all patches are disjoint), and (ii) fill the new patch with items. The second step is performed by using Metropolis-Hastings (MH) (Robert & Casella, 2010) to sample the items that the new patch contains, from the distribution defined in Equation 1. The proposal density we use is defined as follows: (i) add a new item $I_{m+1} = (x_{m+1}, t_{m+1})$ with probability $1/(2P^2 \cdot |\mathcal{T}|)$ (i.e., uniform in position and type), and (ii) remove an existing item $I_i$ with probability $1/2m$ where $m$ is the current number of items in the patch. Before sampling, the patch is initialized by first randomly selecting an existing patch and copying its items into the new patch. This is intended to facilitate rapid mixing of the Markov chain, and reduce the number of MH iterations. Note that if we use small patches and only sample new patches as the agents visit them, boundary effects may be observed due to the missing neighboring patches further away from the agent. For this reason, we actually also sample all missing neighboring patches while sampling each new patch,

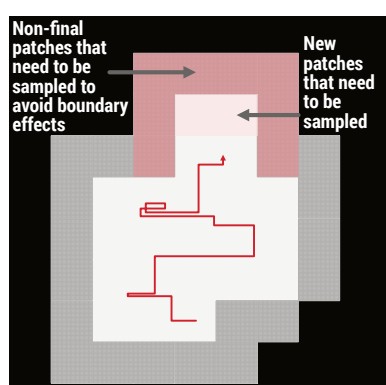

**Figure 2:** Illustration of the procedural generation algorithm for the infinite world map. The $32 \times 32$ patches shown in white have already been sampled and those in gray have been sampled but not fixed in order to avoid boundary effects. The red line corresponds to an example path followed by an agent. Once the agent enters a patch that is not fixed, then that patch is sampled, along with its non-fixed neighboring patches in order to avoid boundary effects.

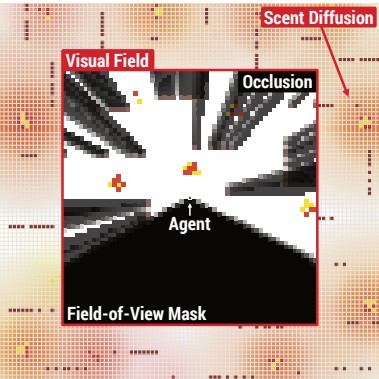

**Figure 3:** Rendering of an agent's perspective from the JBW visualizer.

but do not finalize them (i.e., they are still considered missing and may be resampled later on). This prevents boundary effects during the procedural generation process. An example is shown in Figure 2.

Each item has a color and a scent that is specified by its type and can be perceived by agents. The JBW thus supports two perception modalities, *vision* and *scent*. These modalities are complementary and agents can benefit by learning to combine them, as we explain at the end of this section.

**Vision.** Each agent has a *visual range* property that specifies how far they can see. Vision is represented as a three-dimensional tensor, where the first two dimensions correspond to the width and the height of the agent's visual field, and the third dimension corresponds to the color dimensionality. The visual field is always centered at the agent's current position and the color observed at each cell within the visual field is the sum of the color vectors of all items and agents located at that map location. Agents also have a *field of view* property that specifies their field of view angle (i.e., $180°$ denotes that the agent can only see the forward-facing half of the visual field, whereas $360°$ denotes

that the agent can see the whole visual field). The part of the visual field that is outside an agent's field of view is masked out and appears black to the agent. Another important aspect of vision is that items also have an occlusion property as part of their type. This is used to simulate partial or complete *visual occlusion* (details in Section A.3). If an item with occlusion 1 is in an agent's visual field, then the colors behind that item are not visible to the agent. An example is shown in Figure 3.

**Scent.** Scent is represented as a fixed-dimensional vector, where each dimension can be used to model orthogonal/unrelated scents. Each agent and each item has a pre-specified scent vector that is provided as part of the world configuration (similar to their colors). At each time step, agents can perceive the scent at their current grid position. The physics of scent are described by a simple diffusion difference equation on the world grid. We define the scent at location $(x, y)$ at time $t$ as:

$$S_{x,y}^{t} = \underbrace{C_{x,y}^{t}}_{\text{current items/agents scent}} + \underbrace{\lambda S_{x,y}^{t-1}}_{\text{previous scent}} + \underbrace{\alpha \left( S_{x-1,y}^{t-1} + S_{x+1,y}^{t-1} + S_{x,y-1}^{t-1} + S_{x,y+1}^{t-1} \right)}_{\text{neighboring cells diffused scent}}, \quad (2)$$

where $\lambda$ is the rate of decay of the scent at each location, $\alpha$ is the rate of diffusion of the scent from neighboring grid cells, and $C_{x,y}^{t} = \sum_{I \in \mathcal{I}_{x,y}^{t}} \text{scent}(I) + \sum_{A \in \mathcal{A}_{x,y}^{t}} \text{scent}(A)$, where $\mathcal{I}_{x,y}^{t}$ is the set of all items at time $t$ and location $(x, y)$, and $\mathcal{A}_{x,y}^{t}$ is the set of all agents at time $t$ and location $(x, y)$. Our simulator ensures that the scent (or lack thereof) diffuses correctly, even as items are created, collected, dropped, and destroyed. It does so by keeping track of the creation, collection, drop, and destruction times of each item in the world. Note also that, while simulating this diffusion, we also take into account the non-fixed patches that have been sampled in order to avoid boundary effects.

**Vision-Scent Complementarity.** Vision and scent are complementary. Vision has high precision, in the sense that the agent can see the actual color of each grid cell in its visual field and can thus relatively accurately determine what items may exist in that cell. However, it has low recall—the agent can only see as far as its visual range allows and it has no visual information about the rest of the map. On the other hand, scent has low precision—the scent at the current cell is a linear combination of the scents of all items in the world and it may be very difficult to learn to interpret and use it effectively. However, scent has high recall—the scent at the current cell contains information about items in a much larger range. Thus, learning to use both modalities will be beneficial to agents. In Section 4, we also provide some experimental results supporting this argument.

**Constraints.** The simulator enforces multiple constraints on the actions that agents are allowed to take. We have designed the following small set of constraints with the goal of providing a computationally efficient way to support arbitrarily complex tasks and learning problems:

- Agent Collision: This occurs when multiple agents attempt to move to the same location at the same time. This conflict can be resolved in one of three ways: (i) allow multiple agents to occupy the same location, (ii) first-come-first-serve (only allow the first agent who made a move request for that location to actually move—this is the current default), or (iii) randomly choose one of the agents and satisfy their request (ignoring the requested action of the others).
- Item Blocking Movement: Item types may specify that they block agent movement (e.g., a `Wall` item type). This means that agents are not allowed to move to locations with items of that type.
- Item Collection Requirements: Item types may specify that in order to collect items of that type, an agent has to have first collected a specified number of other items (e.g., collecting `Wood` may only be allowed if the agent has first collected an `Axe`).
- Item Collection Costs: Similar to the collection requirements, item types may specify that in order to collect items of that type, an agent has to drop or destroy a specified number of other items (e.g., collecting an `Axe` may require destroying a piece of `Metal` and a piece of `Wood` that the agent has previously collected).

**Interface.** Users interact with the simulator programmatically. JBW provides functions to add or remove agents from the world, query the current vision and scent perception of each agent, and to direct agents to perform actions. Users can choose to add multiple agents to the world, thus enabling experimentation with multi-agent settings. Multi-agent interactions provide another controllable source of complexity in the JBW. Users can then request actions for each agent in the simulation (i.e., turn, move, do nothing, etc.). Once all agents have requested actions, the simulator executes these actions and advances time, appropriately updating the state of the world.

**Server/Client Support.** The JBW also provides a TCP server-client interface where the simulator can be setup to run as a server. Users (i.e., clients) can then connect to the server, and interact with the simulator by sending messages to the server. This allows use cases such as a class setting where

students can each control an agent in a common simulated world, or perhaps a hackathon where participants can compete in a common world. It also allows debugging and visualization tools to be attached and detached to and from running simulator instances, without affecting the simulations. In fact, this is how our visualizer, which is described in Section 2.3, communicates with the simulator.

**Persistence.** Simulations in the JBW can be saved to and loaded from files, which can then be distributed across platforms. This facilitates *reproducibility*. The simulator guarantees uniform random number generation behavior across all platforms and machines (e.g., in distributed settings). The state of the pseudorandom number generator is also saved and loaded along with the simulation.

## 2.2 ENVIRONMENT

Environments manage simulator instances and provide an interface for performing reinforcement learning experiments using the JBW. We provide implementations of the JBW environments for OpenAI Gym (Brockman et al., 2016) in Python and for Swift RL (Platanios, 2019) in Swift. JBW environments support *batching* by design, with support for parallel execution of the multiple simulator instances being managed (i.e., one simulator for each batch entry). Perhaps the most important aspect of JBW environments is that they require the user to specify a *reward schedule* to use for each experiment. This schedule effectively defines the tasks that the agents are learning to perform. A reward schedule provides a function that, given a simulation time, returns a *reward function* to use at that time. A reward function returns a scalar reward value, given the current and previous states of the agent and the world (e.g., the world map).[1] We provide a simple domain-specific language (DSL) for composing and combining multiple reward functions in arbitrary ways, to allow for the design of composable learning tasks. This enables endless possibilities in the realms of multi-task learning, curriculum learning, and more generally never-ending learning. Currently environments are limited to single agent reinforcement learning settings, but we plan to support multi-agent settings in the future (this is easy because the JBW simulator already supports multiple agents for each simulation).

## 2.3 VISUALIZER

Visualization can be instrumental when developing, debugging, and evaluating never-ending learning systems. To this end, we have implemented a real-time visualizer using Vulkan[2] in which the user can see any part of the simulated JBW, at any scale and simulation rate. The visualizer utilizes the simulator server-client interface to visualize simulations running in different processes or on remote servers, in a fully asynchronous manner. Rendering is multithreaded to provide a smooth and responsive user interface. Finally, the visualizer can be attached, detached, and re-attached to existing simulation server instances, without affecting the running simulations.

## 3 LEARNING TASKS

Learning tasks can be defined in terms of *reward functions* and *reward schedules*, which were defined in Section 2.2. The JBW allows researchers to easily define their own reward functions and schedules, but it also provides a few primitives and ways to compose them in order to effortlessly allow for quick experimentation and prototyping. In fact, all learning tasks used in Section 4 were defined using these primitives. The currently supported primitives are:

| Reward Functions | | Reward Schedules | |
|---|---|---|---|
| `Action[v]` | Give $v$ to agents when they take an action (i.e., not a no-op). | `Fixed[r]` | The reward function is always fixed to $r$, and is thus stationary. |
| `Collect[i,v]` | Give $v$ to agents for each item of type $i$ that they collect. | `Curriculum[{r_i,t_i}_{i=1}^R]` | Use reward function $r_1$ for the first $t_1$ steps, then $r_2$ for $t_2$ steps, ..., and keep using $r_R$ after the list of reward functions is exhausted. |
| `Explore[v]` | Give $v$ to agents each time they move further away from their starting position in the world map. | `Cyclical[{r_i,t_i}_{i=1}^R]` | Use reward function $r_1$ for the first $t_1$ steps, then $r_2$ for $t_2$ steps, ..., and then repeat after the list of reward functions is exhausted. |

We note that `Collect` is a sparse reward function, whereas `Action` and `Explore` are not. For conciseness, we omit the $v$ argument in reward functions when it is set to 1 and we also define `Avoid[i,v]=Collect[i,-v]`.

| Reward Function Compositions | |
|---|---|
| `Combined[r_1,r_2]`
$r_1 \wedge r_2$ | Applies both $r_1$ and $r_2$ and returns the sum of their rewards. |

---

[1]A simple reward function could be one that gives the agent 1 reward point for each `JellyBean` it collects.
[2]Information on Vulkan can be found at `https://www.khronos.org/vulkan/`.

## 4 EXPERIMENTS

The goal of this section is to show how the non-episodic, non-stationary, multi-modal, and multi-task aspects of the JBW make it a challenging environment for existing machine learning algorithms, through a few example case studies. For all experiments we use the simulator configuration and item types shown in Tables 2 and 3.[3] Due to space, the case studies focus on the single-agent setting. We use different agent models depending on which modalities are used in each experiment. If vision is used, then the visual field is passed through a convolution layer with stride 2, $3 \times 3$ filters, and 16 channels, and another one with stride 1, $2 \times 2$ filters, and 16 channels. The resulting tensor is flattened and passed through a dense layer with size 512. If scent is used, then the scent vector is passed through two dense layers: one with size 32, and one with size 512. If both modalities are being used, the two hidden representations are concatenated. Finally, the result is processed by a Long Short-Term Memory (LSTM) network (Hochreiter & Schmidhuber, 1997) which outputs a value for the agent's current state, along with a distribution over actions. Learning is performed using Proximal Policy Optimization (PPO); a popular on-policy reinforcement learning algorithm proposed by Schulman et al. (2017). The experiments are implemented using Swift for TensorFlow.[4]

### 4.1 CASE STUDIES

For all experiments we evaluate performance using the *reward rate* metric, which is defined as the amount of reward obtained per step, computed over a moving window. The size of that window varies per experiment and is reported together with the results. This is an appropriate metric for this task as we want to measure the improvement in the ability of an agent to learn (i.e., the gradient of the reward rate), while also making sure the agent does not get stuck (i.e., the reward rate goes to zero). Whenever possible, we also report the results obtained by the greedy vision-based agent described in Section A.7 of our appendix. The greedy agent makes additional assumptions about the world, and thus, doesn't generalize to more complex environments, it provides a lower bound on the optimal reward rate. Note that a perfect upper bound cannot be obtained as

**Table 2:** Simulator configuration.

| | | |
|---|---|---|
| **Map** | Scent Dimensionality | 3 |
| | Color Dimensionality | 3 |
| | Patch Size | $64 \times 64$ |
| | MH Sampling Iterations | $10,000$ |
| | Scent Decay ($\lambda$) | $0.4$ |
| | Scent Diffusion ($\alpha$) | $0.14$ |
| **Agent** | Color | ■ $[0.00, 0.00, 0.00]$ |
| | Scent | ■ $[0.00, 0.00, 0.00]$ |
| | Action Space | `MoveForward`, `TurnLeft`, and `TurnRight` |
| | Visual Range | 8 |
| | Field-of-View | *experiment-specific* |

**Table 3:** Item types. See Section A.2 for details on the functional forms of the intensity and interaction functions.

| `JellyBean`: Jelly beans appear close to bananas. | |
|---|---|
| Scent | ■ $[1.64, 0.54, 0.40]$ |
| Color | ■ $[0.82, 0.27, 0.20]$ |
| Occlusion | $0.0$ |
| Blocks Agents | `False` |
| Intensity | `Constant[1.5]` |
| Interactions | `JellyBean:PiecewiseBox[10,100,0,-6]` `Banana   :PiecewiseBox[10,100,2,-100]` `Wall     :PiecewiseBox[50,100,-100,-100]` |

| `Banana`: Bananas appear close to jelly beans and away from walls. | |
|---|---|
| Scent | ■ $[1.92, 1.76, 0.40]$ |
| Color | ■ $[0.96, 0.88, 0.20]$ |
| Occlusion | $0.0$ |
| Blocks Agents | `False` |
| Intensity | `Constant[1.5]` |
| Interactions | `JellyBean:PiecewiseBox[10,100,2,-100]` `Banana   :PiecewiseBox[10,100,0,-6]` `Wall     :PiecewiseBox[50,100,-100,-100]` |

| `Onion`: Onions appear scattered all over the world. | |
|---|---|
| Scent | ■ $[0.68, 0.01, 0.99]$ |
| Color | ■ $[0.68, 0.01, 0.99]$ |
| Occlusion | $0.0$ |
| Blocks Agents | `False` |
| Intensity | `Constant[1.5]` |
| Interactions | None |

| `Wall`: Walls tend to be contiguous and axis-aligned. | |
|---|---|
| Scent | ■ $[0.00, 0.00, 0.00]$ |
| Color | ■ $[0.20, 0.47, 0.67]$ |
| Occlusion | 1.0 in experiments with occlusion, 0.0 otherwise |
| Blocks Agents | `True` |
| Intensity | `Constant[-12]` |
| Interactions | `Wall     :Cross[20,40,8,-1000,-1000,-1]` |

| `Tree`: Trees cluster together in irregular shapes. | |
|---|---|
| Scent | ■ $[0.00, 0.47, 0.06]$ |
| Color | ■ $[0.00, 0.47, 0.06]$ |
| Occlusion | 0.1 in experiments with occlusion, 0.0 otherwise |
| Blocks Agents | `True` |
| Intensity | `Constant[2]` |
| Interactions | `Tree     :PiecewiseBox[100,500,0,-0.1]` |

| `Truffle`: Truffles appear in forests and are very rare. | |
|---|---|
| Scent | ■ $[8.40, 4.80, 2.60]$ |
| Color | ■ $[0.42, 0.24, 0.13]$ |
| Occlusion | $0.0$ |
| Blocks Agents | `False` |
| Intensity | `Constant[0]` |
| Interactions | `Truffle  :PiecewiseBox[30,1000,-0.3,-1]` `Tree     :PiecewiseBox[4,200,2,0]` |

---

[3]An example non-stationary world configuration is also presented in Section A.5 of our appendix.
[4]https://www.tensorflow.org/swift.

that would require solving an NP-hard discrete optimization problem.

**Case Study #1: Non-Episodic.** The goal of this case study is to show that the JBW allows for experimenting with never-ending learning agents and to also show how current machine learning methods (e.g., PPO used to train an LSTM-based agent) are failing to effectively perform never-ending learning. For this experiment we use the fixed reward function `Collect[JellyBean]∧ Avoid[Onion]` and let agents interact with the JBW for 10 million steps. Our results are shown in Figure 4. The agents seem to be learning effectively for the first 1 million steps, but start to underperform later on, eventually getting stuck and being unable to collect any reward. This is the case for multiple different learning agents that we

**Figure 4:** Non-episodic experiment result. The reward rate is computed using a 100,000-step window and the shaded bands correspond to standard error over 20 runs.

experimented with; both using different models and using different learning algorithms, such as Deep Q-Networks (DQNs) proposed by Mnih et al. (2013). After connecting the visualizer to observe what happens we see that all agents either: (i) get stuck in an area of the map that they have already explored and exhausted of jelly beans, or (ii) get stuck constantly rotating and not moving to new grid cells at all. This indicates that the JBW is indeed challenging for current machine learning methods when it comes to never-ending learning. Perhaps some sort of reward shaping or curriculum learning could help the agents. However, our goal with this paper is not to solve these hard problems but rather point them out and show how the JBW provides a testbed with which to tackle them.

**Case Study #2: Non-Stationary.** The goal of this case study is to demonstrate that the JBW allows for experimenting with non-stationary and multi-task learning problems. To this end, we perform two experiments: (i) one using a cyclical/periodic reward function schedule where every 100,000 steps we alternate between the `Collect[JellyBean]∧Avoid[Onion]` and `Avoid[JellyBean] ∧Collect[Onion]` reward functions, and (ii) one testing a couple of curriculum reward schedules for eventually learning to `Collect[JellyBean]∧Avoid[Onion]`. The results are shown in Figure 5 and we observe that current standard machine learning approaches are not able to efficiently alternate between different learning problems and are effectively learning each problem from scratch whenever they switch, eventually ending up unable to learn either one. We also observe that agents who first learn to collect jelly beans and then switch to the full reward function are able to learn to collect jelly beans and avoid onions faster than agents that first learn to avoid onions or face the final learning problem directly from the beginning. Eventually all agents perform similarly, but this showcases how the JBW enables research in curriculum learning.

**Case Study #3: Multi-Modal.** The goal of this case study is to: (i) show how computationally efficient features, such as the field of view mask and visual occlusion, allow for increasing the learning problem complexity in a controllable manner and, perhaps most importantly, (ii) show how the perception modalities of the JBW are complementary. We thus perform three experiments. For the first two we use the fixed reward function `Collect[JellyBean]` and for the last one we use `Collect[Onion]`. We change the reward function in order to show how easy it is to experiment with different tasks in the JBW. In the first experiment, we vary the field of view of the agents. The results are shown in the left plot of Figure 6. We see that decreasing the field of view allows us to make the learning task harder for agents, while maintaining the same computational cost for the environment. Similarly, in the second experiment we measure the effect that visual occlusion has on performance. The results are shown in the middle plot of Figure 6 and we observe that enabling visual occlusion makes the learning task harder. Finally, with the third experiment our goal is to show that vision and scent are complementary. The results are shown in the right plot of Figure 6. We see that "vision" agents do better than "scent" agents, indicating that vision is perhaps an easier perception modality to use in the context of this learning task. Surprisingly though, the "vision" agents also do better than the "vision+scent" agents. This indicates a limitation of the model because, even though scent contains useful information that vision does not, the agents seem to get confused by it and do not seem able to use it properly. It also shows the need for better multi-modal algorithms and the utility of the JBW in testing such algorithms. The relative utility of scent and vision depends on the environment and the agent model. We demonstrate this in Section A.4 by providing a different configuration where "scent" agents are able to outperform both "vision" and "vision+scent" agents.

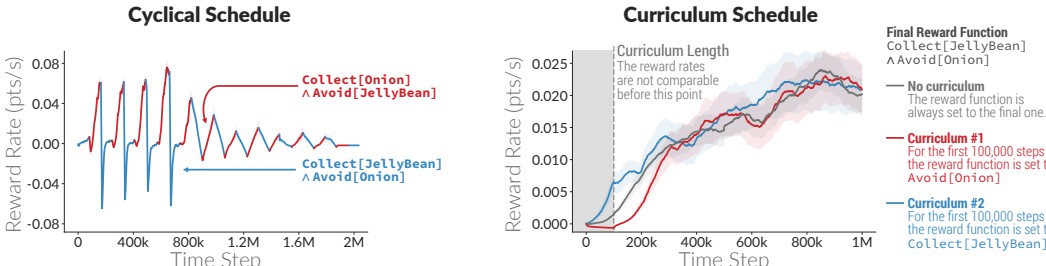

**Figure 5:** Non-stationary experiment results. The reward rate is computed using a 100,000-step window and the shaded bands correspond to standard error over 20 runs.

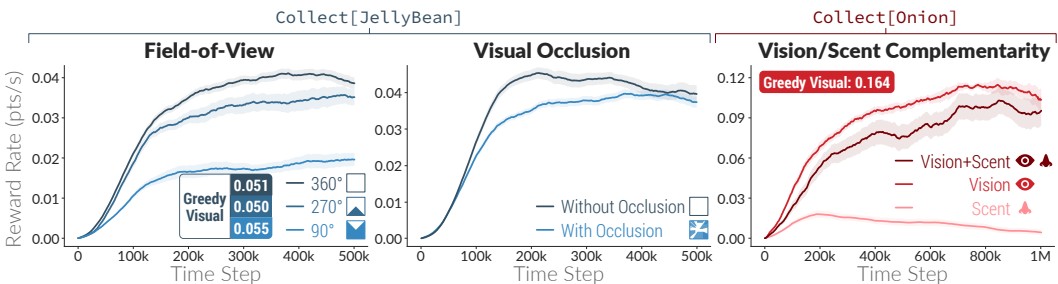

**Figure 6:** Multi-modal experiment results. The braces on top of the plots specify the reward function used in each case. The reward rate is computed using a 100,000-step window and the shaded bands correspond to standard error over 20 runs. "Greedy Visual" refers to the reward rate obtained by the greedy visual agent baseline described in Section A.7 (which, as explained in that section, is not able to handle occlusion or avoidance).

## 5 CONCLUSION AND FUTURE EXTENSIONS

We presented a new testbed designed to facilitate experimentation with never-ending learning agents, where the *complexity* of the learning problems is higher than that of existing testbeds and evaluations, while maintaining controllability, performance, and reproducibility. In order to produce more complex environments, the JBW supports non-stationary environments, with multiple distinct but inter-related tasks and complementary perception modalities. The JBW also explicitly restricts learning to a single never-ending episode. It is highly configurable and performant, and provides users with tools to easily save, load, distribute environments, and reproduce and visualize results. We also showed how easily we can define learning tasks in the JBW, for which current machine learning methods struggle.

The space of potential extensions to the JBW is large. Although the current intensity and interaction functions are stationary with respect to space (i.e., they are independent of position $x, y$), it is not difficult to define new non-stationary functions, in order to generate worlds with non-stationary item distributions. The JBW supports multiple agents running asynchronously in the world, and so it would also be interesting to experiment with multi-agent settings. However, agents currently don't have an easy way to communicate with each other, and so adding a mechanism for communication, perhaps via new agent-item interactions (e.g., reading/writing note items), would be interesting. Another way to add complexity is via items that can contain "strings" (e.g., notes) in an agent-specific language, or even natural language. These notes could, for example, contain task specifications. Scent currently does not interact with items in the world, meaning that it can pass through Wall items without any hinderance. Thus, another possible extension would be to support more complex item-scent interactions. It would be interesting to explore interactions between items and the properties of agents (e.g., a Telescope could extend the visual range of an agent while narrowing its field of view). Finally, another interesting way to add complexity is to generate richer relationships between item types, perhaps even an ontology. We look forward to continue improving the JBW, and hope that a standardized testbed for never-ending learning will motivate research into more generally-intelligent learning algorithms.

### ACKNOWLEDGMENTS

We thank the anonymous reviewers for their helpful comments and the Spring 2018 class in "Never-Ending Learning" at Carnegie Mellon University for the impulse and discussion that led to the development of this framework. This research was supported in part by AFOSR grant FA95501710218.

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

## A APPENDIX

### A.1 COMPLEXITY BOUND

We show that $K(\mathcal{E})$ must be at least $K(\pi^*)$ up to a constant. We can write an algorithm $\hat{\pi}$ that enumerates all possible sequences of environment states, observations, actions, and rewards from time $t$ up to time $T$: $(s_t, \omega_t, a_t, r_t), \ldots, (s_T, \omega_T, a_T, r_T)$. Then $\hat{\pi}$ computes the action that maximizes the expected reward $\mathbb{E}[\mathcal{R}(\{r_k\}_{k=t}^T)]$. Since the images $\text{im}(\hat{\pi}) \subseteq \mathcal{A}$ and $\text{im}(\pi^*)$ are discrete, and $\lim_{T\to\infty} \arg\max_{a_t} \mathbb{E}[\mathcal{R}(\{r_k\}_{k=t}^T)] = \arg\max_{a_t} \mathbb{E}[\mathcal{R}(\{r_k\}_{k=t}^\infty)]$, there is a sufficiently large finite $T$ such that the action computed by $\hat{\pi}$ is the same as that computed by $\pi^*$. Note that $\hat{\pi}$ relies on a subroutine that simulates the environment $\mathcal{E}$ in order to first enumerate the environment states, and the subroutine to perform the optimization is independent of $\mathcal{E}$, and so $K(\hat{\pi}) = K(\mathcal{E}) + c$ for a constant $c$. Since $K(\pi^*) \leq K(\hat{\pi})$, we have that $K(\mathcal{E}) \geq K(\pi^*) - c$.

### A.2 INTENSITY AND INTERACTION FUNCTIONS

The JBW currently only supports a small number of implemented intensity and interaction functions to control the distribution of items in the world. However, it is very straightforward to implement new customized intensity and interaction functions. Let $(x, y) \in \mathbb{Z}^2$ be a position and $t \in \mathcal{T}$ be an item type. Intensity functions are indexed by item type, and so each item type is assigned its own intensity function: $f((x, y), t) \triangleq f_t(x, y)$. The JBW currently supports three intensity functions:

1. `Zero`: $f_t(x, y) = 0$.
2. `Constant[v]`: $f_t(x, y) = v$.

3. `RadialHash[`$\Delta$`,`$s$`,`$c$`,`$k$`]`:

$$f_t(x, y) = c - k \cdot \hat{M}(\sqrt{x^2 + y^2}/s + \Delta),$$

where $\hat{M} : \mathbb{R} \mapsto [0, 1]$ is the linear interpolation of $M(\lfloor t \rfloor)/(2^{32} - 1)$ and $M(\lfloor t + 1 \rfloor)/(2^{32} - 1)$, and $M : \mathbb{Z} \mapsto \mathbb{Z}$ is the last mixing step of the 32-bit MurmurHash function (Appleby, 2008). This provides pseudorandomness to the intensity function.

For interaction functions, let $(x_1, y_1)$ be the input position of the first item, $t_1$ be the type of the first item, $(x_2, y_2)$ be the position of the second item, and $t_2$ be the type of the second item. Interaction functions are indexed by pairs of item types, so each pair of item types can be given its own interaction function: $g(((x_1, y_1), t_1), ((x_2, y_2), t_2)) \triangleq g_{t_1, t_2}((x_1, y_1), (x_2, y_2))$. The JBW currently supports four interaction functions:

1. `Zero`: $g_{t_1, t_2}((x_1, y_1), (x_2, y_2)) = 0$.
2. `PiecewiseBox[`$U$`,`$V$`,`$u$`,`$v$`]`:

$$g_{t_1, t_2}((x_1, y_1), (x_2, y_2)) = \begin{cases} u, & \text{if } d < U, \\ v, & \text{if } U \leq d < V, \\ 0, & \text{otherwise}, \end{cases}$$

where $d = (x_1 - x_2)^2 + (y_1 - y_2)^2$.
3. `Cross[`$U$`,`$V$`,`$u$`,`$v$`,`$\alpha$`,`$\beta$`]`:

$$g_{t_1, t_2}((x_1, y_1), (x_2, y_2)) = \begin{cases} u, & \text{if } d = 0, D \leq U, \\ \alpha, & \text{if } d \neq 0, D \leq U, \\ v, & \text{if } d = 0, U < D \leq V, \\ \beta, & \text{if } d \neq 0, U < D \leq V, \\ 0, & \text{otherwise}, \end{cases}$$

where $d = \min\{|x_1 - x_2|, |y_1 - y_2|\}$ and $D = \max\{|x_1 - x_2|, |y_1 - y_2|\}$.
4. `CrossHash[`$s$`,`$c$`,`$k$`,`$\delta$`,`$u$`,`$v$`,`$\alpha$`,`$\beta$`]` is the same as `Cross[`$U$`,`$V$`,`$u$`,`$v$`,`$\alpha$`,`$\beta$`]` except for the fact that $U$ and $V$ are given by:

$$U = c + k \cdot \hat{M}(x_1/s),$$
$$V = U + \delta,$$

where $\hat{M}(\cdot)$ is defined as above in the `RadialHash` intensity function.

Even though this is a small set of intensity and interaction functions it can allow for creating worlds with many interesting features (e.g., we use the `Cross` interaction function to create contiguous wall segments that are axis-aligned, and the `PiecewiseBox` interaction function to create irregularly shaped clusters of trees forming forests). Note that the unspecified intensity and interaction functions in Table 3 are set to `Zero` by default.

## A.3 FIELD OF VIEW AND VISUAL OCCLUSION

To compute the color of a cell with respect to the agent's field of view, let the cell position be $(x, y)$ and consider a circle of radius $\frac{1}{2}$ centered at $(x, y)$. Project this circle onto the circle of radius 1 centered at the agent position. Let $\theta$ denote this projection (an arc). Let $\theta_{\text{fov}}$ be the arc on the agent's circle centered on a point in the current agent direction. The length of $\theta_{\text{fov}}$ is specified by the field-of-view parameter in the configuration. The color of cell $c_{x,y}$ is then computed as:

$$c_{x,y} = \hat{c}_{x,y} \cdot \frac{|\theta_{\text{fov}} \cap \theta|}{|\theta|}, \tag{3}$$

where $\hat{c}_{x,y}$ is the original color of the cell. In order to compute how much a cell at position $(x, y)$ is occluded, we consider a circle of radius $\frac{1}{2}$ centered at $(x, y)$, and project this circle onto the circle of radius 1 centered at the agent position. Let $\theta$ denote this projection (an arc). Each item in the agent's visual field is similarly projected onto the agent's circle, each producing an arc $\theta_i$. The color of the cell $c_{x,y}$ is then computed as:

$$c_{x,y} = \hat{c}_{x,y} \cdot \max\left\{1 - \sum_i o_i \frac{|\theta_i \cap \theta|}{|\theta|}, 0\right\}, \tag{4}$$

where $\hat{c}_{x,y}$ is the original color of the cell, and $o_i$ is the occlusion parameter of the $i^{th}$ item, as specified by the item's type. If a cell is affected by both the field of view and visual occlusion, the above effects are composed (both multiplicative factors are applied to the original color).

## A.4 RELATIVE UTILITY OF SCENT AND VISION

The relative difficulty of utilizing scent or vision to effectively navigate in the JBW environments depends on the environment configuration as well as the method used by the learning agent. For example, if the learning agent does not possess memory, it cannot remember the scent of any previously visited tile, and thus it will not be able to determine the direction from which the scent is diffusing. Therefore, such methods will not benefit from the information provided by the scent modality. In addition, scent is not necessarily blocked by items that block movement (e.g., walls). Thus, in environments with such items, scent is more difficult to utilize, since the simple strategy of following paths of monotonically increasing scent could lead to a wall. The agent could be fooled to believe an item is nearby when, in fact, it is behind a wall. This is illustrated in Figure 7 and is possibly one of the main reasons the "vision+scent" agent underperforms the "vision" agent in Figure 6.

In order to showcase how scent can provide useful information, we also designed a simpler environment configuration that does not contain any walls. The configuration for this world is shown in Tables 4 and 5. Given the task of collecting `JellyBeans`, we expect the scent modality to be very useful as long as the agent has some sort of memory. The results of using an LSTM-based agent are shown in Figure 8. We observe that the "scent" agent outperforms both the "vision" and the "vision+scent" agents. This is the opposite pattern of what we observe in Figure 6.[5] This can partly be explained by the fact that, in this case, the visual field of the agent is more restricted, thereby limiting the agent's reliance on vision in its learning. Ideally agents should be able to use each perception modality optimally and not be "confused" by the fact that one of them may be harder to utilize than the other. Thus, these experiments showcase how the JBW can be used to evaluate multi-modal machine learning algorithms.

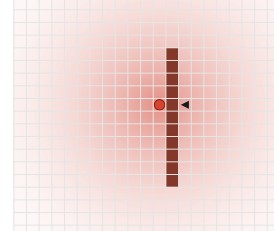

Scent passes through walls and so the agent needs to learn that even though it can smell an item being close, that does not mean the shortest path is feasible and needs to combine the scent information with the visual information.

**Figure 7:** Example showing one of the challenges the scent modality poses for agents.

**Table 4:** Simulator configuration.

| | | |
|---|---|---|
| Map | Scent Dimensionality | 3 |
| | Color Dimensionality | 3 |
| | Patch Size | $32 \times 32$ |
| | MH Sampling Iterations | $4,000$ |
| | Scent Decay ($\lambda$) | 0.4 |
| | Scent Diffusion ($\alpha$) | 0.14 |
| Agent | Color | ■ [0.00, 0.00, 0.00] |
| | Scent | ■ [0.00, 0.00, 0.00] |
| | Action Space | `MoveForward,` `TurnLeft,` and `TurnRight` |
| | Visual Range | 5 |
| | Field-of-View | $60°$ |

**Table 5:** Item types for the simple environment where the scent modality dominates the vision modality. See Section A.2 for details on the functional forms of the intensity and interaction functions.

| `JellyBean`: Jelly beans appear close to bananas. | |
|---|---|
| Scent | ■ [0.0, 0.0, 1.0] |
| Color | ■ [0.0, 0.0, 1.0] |
| Occlusion | 0.0 |
| Blocks Agents | `False` |
| Intensity | `Constant[-5.3]` |
| Interactions | `JellyBean:PiecewiseBox[10,200,0,-6]` `Banana   :PiecewiseBox[10,200,2,-100]` `Onion    :PiecewiseBox[200,0,-100,-100]` |

| `Banana`: Bananas appear close to jelly beans. | |
|---|---|
| Scent | ■ [0.0, 1.0, 0.0] |
| Color | ■ [0.0, 1.0, 0.0] |
| Occlusion | 0.0 |
| Blocks Agents | `False` |
| Intensity | `Constant[-5.3]` |
| Interactions | `JellyBean:PiecewiseBox[10,100,2,-100]` `Banana   :PiecewiseBox[10,200,0,-6]` `Onion    :PiecewiseBox[200,0,-6,-6]` |

| `Onion`: Onions appear scattered, away from jellybeans and bananas. | |
|---|---|
| Scent | ■ [1.0, 0.0, 0.0] |
| Color | ■ [1.0, 0.0, 0.0] |
| Occlusion | 0.0 |
| Blocks Agents | `False` |
| Intensity | `Constant[-5]` |
| Interactions | `JellyBean:PiecewiseBox[200,0,-100,-100]` `Banana   :PiecewiseBox[200,0,-6,-6]` |

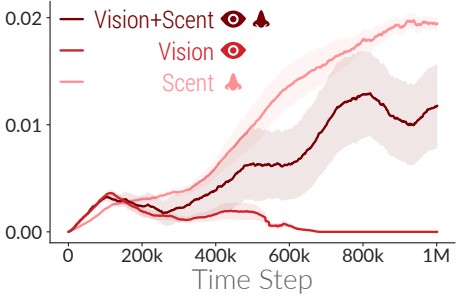

**Figure 8:** Results of experiments on the configuration shown in Tables 4 and 5.

---

[5]Note that since the configurations differ, the reward rates between the two figures are not directly comparable.

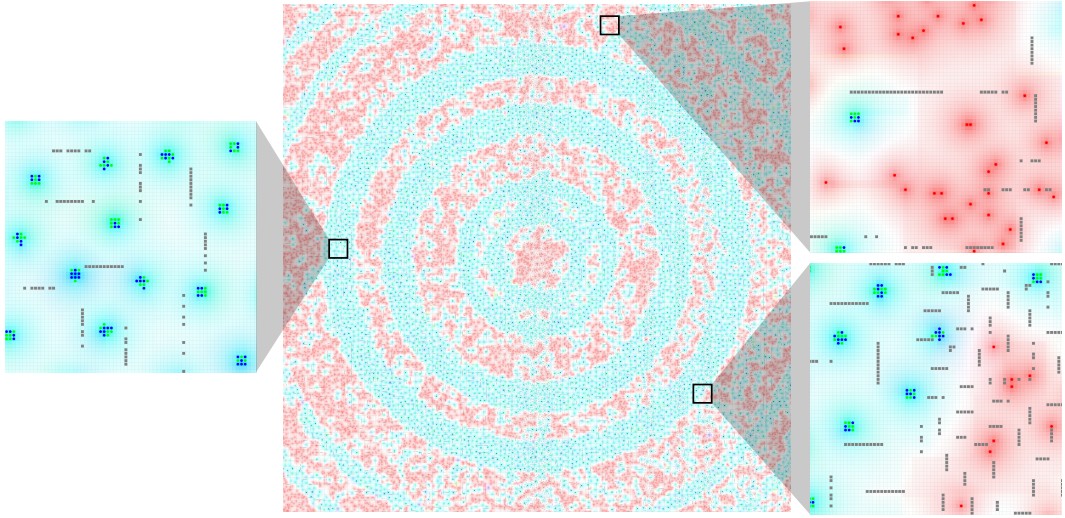

**Figure 9:** Visualization of an example environment with spatial non-stationarity. Each tile is colored according to its scent. `JellyBeans` are shown in blue, `Bananas` in green, and `Onions` in red. `Walls` are depicted as grey squares. This environment was generated using the configuration shown in Tables 6 and 7.

## A.5 EXAMPLE OF SPATIAL NON-STATIONARITY

To demonstrate the ability to generate spatially non-stationary worlds in JBW, we provide a configuration that makes use of non-stationary intensity and interaction functions. The configuration is shown in Tables 6 and 7, and a visualization is provided in Figure 9. `JellyBeans` and `Onions` appear together in clusters and, since this configuration uses the non-stationary intensity function `RadialHash`, these clusters are arranged in concentric circles around the origin that are irregularly spaced. `RadialHash` uses a hash function to induce a pseudorandom relationship between the distance to the origin and the likelihood of finding such clusters. `Walls` in this environment also have a non-stationary distribution. In some regions, they are smaller and more frequent, whereas in other regions they are longer and appear more sporadically.

**Table 6:** Item types for the non-stationary environment. See Section A.2 for details on the functional forms of the intensity and interaction functions.

`JellyBean`: Jelly beans appear close to bananas.

| | |
|---|---|
| Scent | ■ $[0.0, 0.0, 1.0]$ |
| Color | ■ $[0.0, 0.0, 1.0]$ |
| Occlusion | 0.0 |
| Blocks Agents | `False` |
| Intensity | `RadialHash[500,60,-3.0,14]` |
| Interactions | `JellyBean:PiecewiseBox[10,200,0,-6]` `Banana   :PiecewiseBox[10,200,2,-100]` `Onion    :PiecewiseBox[200,0,-100,-100]` |

`Banana`: Bananas appear close to jelly beans.

| | |
|---|---|
| Scent | ■ $[0.0, 1.0, 0.0]$ |
| Color | ■ $[0.0, 1.0, 0.0]$ |
| Occlusion | 0.0 |
| Blocks Agents | `False` |
| Intensity | `RadialHash[500,60,-3.0,14]` |
| Interactions | `JellyBean:PiecewiseBox[10,100,2,-100]` `Banana   :PiecewiseBox[10,200,0,-6]` `Onion    :PiecewiseBox[200,0,-6,-6]` |

`Onion`: Onions appear scattered, away from jellybeans and bananas.

| | |
|---|---|
| Scent | ■ $[1.0, 0.0, 0.0]$ |
| Color | ■ $[1.0, 0.0, 0.0]$ |
| Occlusion | 0.0 |
| Blocks Agents | `False` |
| Intensity | `Constant[-5]` |
| Interactions | `JellyBean:PiecewiseBox[200,0,-100,-100]` `Banana   :PiecewiseBox[200,0,-6,-6]` |

`Wall`: Walls tend to be contiguous and axis-aligned.

| | |
|---|---|
| Scent | ■ $[0.0, 0.0, 0.0]$ |
| Color | ■ $[0.5, 0.5, 0.5]$ |
| Occlusion | 1.0 in experiments with occlusion, 0.0 otherwise |
| Blocks Agents | `True` |
| Intensity | `Constant[0]` |
| Interactions | `Wall     :CrossHash[60,4,25,2,20,-200,-20,1]` |

**Table 7:** Simulator configuration.

| | | |
|---|---|---|
| Map | Scent Dimensionality | 3 |
| | Color Dimensionality | 3 |
| | Patch Size | $32 \times 32$ |
| | MH Sampling Iterations | $4,000$ |
| | Scent Decay ($\lambda$) | 0.4 |
| | Scent Diffusion ($\alpha$) | 0.14 |
| Agent | Color | ■ $[0.00, 0.00, 0.00]$ |
| | Scent | ■ $[0.00, 0.00, 0.00]$ |
| | Action Space | `MoveForward`, `TurnLeft`, and `TurnRight` |
| | Visual Range | 5 |
| | Field-of-View | *experiment-specific* |

## A.6 PERFORMANCE

The JBW is implemented in optimized C++, with performance being highly prioritized in both its design and its implementation. This would allow less time and hardware resources to be spent simulating the world and more time and resources to be allocated for the machine learning algorithms. Additionally, the JBW is perceptually quite simple, being a two-dimensional grid world with limited vision and scent inputs. This allows the machine learning algorithm to focus less on perceptual information processing and more on abstract information processing, which we think is a hallmark of never-ending learning. As a rough indication of performance, on a single core of an Intel Core i7 5820K (released in 2014) at 3.5GHz, the JBW can generate $8.56$ patches per second, each of size $64 \times 64$ (i.e., $35,062$ grid cells), using the configuration described in Section 4.

## A.7 GREEDY ALGORITHM

As a benchmark and for the sake of comparison, we also implemented a simple greedy agent that searches its visual field for cells of a particular color, and then computes the shortest path to those cells. This algorithm makes the assumption that reward is maximized simply by collecting items of a single color, ad infinitum. It also assumes that this color is known a priori. Additionally, it assumes the color of obstacles (items that block agent movement or that should be avoided as part of the reward function) is known a priori, and is distinct from the color of items that provide reward. The shortest path it computes is such that it never goes through any such obstacles.

---

**Algorithm 1:** Pseudocode for the greedy vision-based algorithm.

**Input:** Color of rewarding items $c_r$ and color of obstacles $c_w$.

1   Initialize `best_path = null`
2   **Function** `GreedyVisionPolicy` (*visual field $\omega_t$*)
3      `shortest_path = ShortestPath`$(\omega_t, c_r, c_w)$
4      **if** *best_path = null or $|shortest\_path| < |best\_path|$*
5         assign `best_path = shortest_path`
6      **if** *best_path = null*
7         **if** *the cell immediately in front of the agent has color $\gamma c_w$ for any $\gamma > 0$*
8            **return** *MoveForward*
9         **else**
10           **return** *TurnLeft or TurnRight uniformly at random*
11      **else**
12         `next_action` = dequeue the next action from `best_path`
13         **if** *best_path has no further actions*
14            assign `best_path = null`
15         **return** *next_action*

---

The algorithm is shown in pseudocode in Algorithm 1. The function `ShortestPath` is simply Dijkstra's algorithm on a directed graph where each vertex corresponds to a unique agent position and direction within its visual field $\omega_t$, and each edge corresponds to a possible action that transitions between agent states (Dijkstra, 1959). Let $c_r$ be the color of items that provide reward, and $c_w$ be the color of items that block agent movement. The algorithm returns a shortest path from the agent's current position and direction to a cell that has a color $\gamma c_r$ for any $\gamma > 0$, while avoiding cells that have color $\gamma c_w$ for any $\gamma > 0$ (we match any color in the direction of the vectors $c_r$ and $c_w$ in order to detect partially occluded items). If no such path exists, `ShortestPath` returns `null`. In the case where the agent's field of view is limited, `ShortestPath` only returns paths that pass through cells within the agent's field of view. Also, in the experiment where the agent must additionally avoid `Onion` items, `ShortestPath` avoids them in the same way that it avoids obstacles: it avoids cells that have color $\gamma c_o$ for any $\gamma > 0$ where $c_o$ is the color of the `Onion` item type.

In environments with visual occlusion, if items with high occlusion are arranged in a line (such as a wall), and the agent is adjacent to the wall and facing it, the portions of the wall further from the agent will be occluded by the portion of the wall closer to the agent. Since we currently do not distinguish between empty cells and completely occluded cells, `ShortestPath` will return paths that may pass through the wall. If no other paths are returned, the agent will continuously try to move through the wall and make no progress.

Similarly, in environments containing items that provide negative reward, the above greedy algorithm does not allow the agent to move over tiles with such items. Thus, given sufficient time, the agent will find itself moving between two clusters of avoided items, where one cluster is shaped like a "∪" and the other like a "∩". And so when the agent moves downward into the "∪" of one cluster, it will make a 180-degree turn, leaving the cluster, and moving upward toward the other cluster. The other cluster is shaped symmetrically, and the agent will repeat the behavior deterministically: enter the "∩", make a 180-degree turn, leave the cluster, and move down toward the first cluster again. This will repeat indefinitely.

