# OpenReview forum: "Jelly Bean World: A Testbed for Never-Ending Learning"
_ICLR.cc/2020/Conference — Accept (Poster)_

### Official Review · AnonReviewer1 · 2019-10-19
**Official Blind Review #1**

**Rating:** 6

**Review:**

Summary

This paper introduces a new environment for testing lifelong or never-ending learning. The goal of the environment is to act as a new benchmark testbed for challenging existing agents and models across areas of research, encouraging and pushing new research towards solving challenges in curriculum learning, exploration, representation learning, and continual learning. The contributions in this paper extend upon previous work by building an easily controllable environment generator with key necessary features for lifelong learning including: non-stationarity, multiple task specification, and multiple sets of observable features.

Review

The paper highlights many key characteristics of an environment that are challenging to current RL models. This focus on building a benchmark upon which further research can measure performance is important. I find the proposed environment to be incredibly intriguing and would find it valuable to the field of lifelong learning (or continual learning or never-ending learning, etc.). I think the size and scope of the environment generator is impressive, showing a considerable amount of engineering effort has gone into its design.

The largest overarching issue that I would like to point out is the limited study of modelling choices. I am not an expert on applied Reinforcement Learning, so I can make very few claims about the validity of the chosen network architecture or use of the PPO policy-gradient algorithm for this environment. However, it is critical, in my view, that a paper introducing a new environment studies these effects itself; demonstrating how various degrees of learning capacity or wider ranges of learning algorithms behave in the given environment. If a slightly larger network architecture trivially solves each task in this environment, can this still be considered a benchmark task? A key result in the paper that I would like to see further investigated (even with only a different network architecture) would be Figure 6, the comparison between scent, vision, and vision+scent. It is unclear to me why the scent features would be so challenging to learn from and specifically why they would harm the representation so permanently. A deeper study using only the scent features would be valuable to me. In its current state, it appears that these feature provide no additional information and are thus not necessary to include in the environment; breaking one of the primary motivating features of JBW: the multi-modality. I recognize that the paper comments on the orthogonality of the scent playing a role, and notes that further results are included on a not-yet-available website (presumably to maintain anonymity). However, I would like to see these results included in the appendix of the paper so I could better assess the utility of the scent features. Perhaps an additional result showing the average reward versus the cosine distance (or other measure of orthogonality) between "jellybeans" and "onions" would additionally motivate the utility of the scent features.

The paper empirically investigates the use of curriculum learning to accelerate learning for a particular task. The paper then claims that curriculum learning improves learning speed, but ultimately does improve final performance. This demonstration is intended to showcase the use of the proposed environment (JBW) for curriculum learning. However, there are few key issues with this empirical study. First, the paper shows the reward rate of 3 different curricula but does mention the metric used to compare the agents during the time the curricula is active. It is implied that the metric is the reward rate of each individual agent; however, each agent has a unique reward function making comparisons between agents impossible. Curriculum #2 can only receive positive rewards while Curriculum #1 can only receive negative rewards. Naturally this means that Curriculum #2 must have strictly greater or equal reward rate over Curriculum #1. Even in the case that the final objective specifies the metric used, these are still highly non-comparable entities. A suggestion to improve this result would be to run each curricula for 100k as a "pretraining" phase, then to restart the agents to the same state in the environment and measure their performance from there.

The case study measuring the effects of non-stationarity of the rewards does not provide sufficient evidence that the proposed environment contributes a novel ability to investigate non-stationarity. First, the given study of non-stationarity focuses solely on an alternating reward function, clearly demonstrating the problem of catastrophic forgetting. While this is a motivating demonstration, it is not novel and the issue of catastrophic forgetting in our models has been known since at least the 90s (e.g. French 1999 and related). Carefully and scientifically investigating such an issue is best done in a far less complex environment where more precise results can be drawn. Further, the ability to oscillate a reward function in this way is not unique to this environment and can be trivially done in most environments. Secondly, it is unclear if JBW allows for non-stationarity in the transition probabilities in the MDP. This is a critical component to non-stationarity and would be a necessary feature for me to claim non-stationarity is widely supported in the environment.

The paper starts with a motivating conversation about environment complexity, with interesting insights into measuring the complexity of an environment based on the complexity of the policy used to solve that environment. However this conversation is ignored until the conclusion of the paper, where the paper claims to have built an environment of greater complexity than already existing environments. Without any supporting evidence in the body of the paper, it is impossible to verify the validity of this statement, and it is still an open question to me whether this claim is even falsifiable in the first place. As a concrete counter-claim, I would claim that the Minecraft environment (Malmo) has similar or higher complexity to the proposed environment in most aspects. Minecraft has a far greater diversity of objects, a third dimension of movement, adversarial components, hunger and health, etc. each of which adding a large level of complexity not achievable in the proposed environment. This is not to say that I expect the proposed environment to contain these features, but rather to point out that claims of greater complexity may be ill-founded.

Additional Comments (not affecting score)

I do slightly question if ICLR is the appropriate venue for such work. While I recognize that the scope of this conference has shifted considerably over the past few years, this paper (as written) does not further understanding or study of learning representations. I believe a more careful demonstration of the representation induced by characteristics of the environment is within easy reach of the paper, but is not currently presented.

-----------

After the author response, reading other reviews/responses, and looking at the edited draft:

I am convinced of the utility of the domain, the scope of the engineering effort put into building, and the ease with which it can be configured by the user to test many applicable settings (partial observability, stochasticity in transitions and rewards, etc.). I remain slightly skeptical of the amount of benefit the proposed provides over the Malmo environment for any of the settings discussed in the case-studies.

I specifically feel my concerns about the stochasticity in the transitions and environment complexity have been well addressed. My concerns about the curriculum learning demonstration are partially addressed to a point where I am satisfied. My concerns about the modeling choice are also partially satisfied, with one lingering concern. I am unclear if the environment is trivially solvable by using more computation resources (e.g. bigger networks). However, after reconsideration I decided this concern bares less weight than I previously considered.

All this considered, I am changing my rating from 3 -> 6.

**Experience Assessment:**

I have published one or two papers in this area.

**Review Assessment: Checking Correctness Of Derivations And Theory:**

N/A

**Review Assessment: Checking Correctness Of Experiments:**

I carefully checked the experiments.

**Review Assessment: Thoroughness In Paper Reading:**

I read the paper at least twice and used my best judgement in assessing the paper.

---

> ### Author Response · Authors · 2019-11-10
> **Response to Official Blind Review #1**
>
> First, we thank you for your helpful comments and suggestions. We address them in the order in which they were presented.
>
> 1. Study of Modeling Choices:
>
> We agree that it is important to, at least, demonstrate the utility of scent as an orthogonal feature to vision. Since we were not able to fit these results in the main paper, we have added an appendix section with experimental results with a different environment configuration in which an agent learns to rely on scent as opposed to vision. In this section, we also added a discussion on how the differences in the environment could affect the relative utility of modalities. These results also demonstrate that it is possible to construct worlds in JBW in which agents learn more effectively by using scent than by using vision. In this section, we also discuss how the choice of the learning algorithm (such as the neural architecture) can affect the relative utility of the modalities. We could also vary the architecture as you suggested. We do not present results using different architectures in the paper as varying the size of the architecture did not change the observed patterns in our experiments. We believe that better utilizing and combining the two signals provided by the vision and scent modalities would necessitate better methods for multi-modal information fusion and that is one of the challenges that the JBW provides for future research.
>
> We also ran experiments using different RL algorithms, such as Deep Q-Networks (DQNs) instead of PPO and our initial results were similar to those presented in the paper, with slightly worse overall performance. We have not included these results because we have not run experiments with DQNs for all of the proposed settings due to computational constraints. If you strongly believe doing so will strengthen our paper we are willing to perform a full evaluation using DQNs and add the results in a new appendix section.
>
> 2. Curriculum Learning:
>
> This is a valid point. However, the curriculum only lasts for the first 100k steps and even though reward rates are indeed not comparable during that interval, they are comparable after the first 100k steps and we do observe differences between the three curricula. Thus our observations and key takeaways still hold. We have added a clarification to Figure 5 indicating that the reward rates for the first 100k steps are not comparable.
>
> 3. Non-Stationarity:
>
> First, we would like to emphasize that users of JBW can induce non-stationarity in the MDP transition probabilities by generating environments where the distribution of items is not stationary in space. This can be done by using non-stationary intensity and interaction functions (refer to Eq. 1). Due to page limit constraints, we were not able to add an example of this in the main paper, but we agree that an example of a non-stationary generated environment would be valuable. *To this end, we have added a new appendix section with a configuration that produces a non-stationary environment and a visualization thereof.* Note that non-stationarity can also be induced by multi-agent settings, which the JBW already supports.
>
> Also, please note that the oscillating reward function was only meant to provide a simple demonstration of a reward function that is not stationary in time. Our environment gives users the ability to create arbitrarily complex non-stationary reward schedules that may even depend on the agents’ past actions.
>
> 4. Environment Complexity:
>
> Our main goal is to provide an environment that is not only complex, but also controllable and efficient so that it allows for controlled experiments using a manageable amount of computing power. Note that if complexity was the sole objective, the real-world environments would be ideal. However, real-world environments do not allow us to test for certain properties in isolation and/or control for their complexity. A similar argument can be made about Minecraft. It requires rendering of a 3D world that is computationally expensive, and also requires large models to be trained in order to learn to perceive that world alone (without accounting for the complexity of any given task). Furthermore, Minecraft can be considered stationary (for single-agent experiments—its map is generated based on Perlin noise), the map generation process is not controllable, and the set of available items and their interactions are fixed. Having said that, Minecraft does provide a complex and interesting environment for learning but it lacks the aforementioned features that we consider highly desirable and motivate in the introduction and Table 1.
>
> We revised the first sentence of the conclusion to try to better communicate this point: “We presented a new testbed designed to facilitate experimentation with never-ending learning agents, where the complexity of the learning problems is higher than that of existing testbeds, while maintaining controllability, performance, and reproducibility."

---

### Official Review · AnonReviewer3 · 2019-10-22
**Official Blind Review #3**

**Rating:** 6

**Review:**

Jelly Bean World: A Testbed for Never-Ending Learning
This work introduces a domain for evaluating and experimenting with algorithms for never-ending learning. These are variants of grid worlds which have multi-task, multi-modal, dynamic settings and can lead to interesting learning challenges.

I’m referring to never-ending learning as NEL throughout.

Introduction
Comment: It’s good to tell the reader as early as possible why never ending learning is different than multi-task or continual or lifelong learning? Those are more commonly used terms in the community so it should be situated properly. All of this stuff about NEL seems very similar to continual learning/lifelong learning and we should really reference it and describe the difference?

I didn’t find “In order to more formally describe general intelligence, we posit that there is an underlying measure of complexity of the environment E such that: (i) highly specialized and non-general learning algorithms can perform well in environments with low complexity, but (ii) environments with high complexity require successful learning agents to possess more general learning capabilities” to provide much clarity. Can we either remove it or stick to the later formalism?

In never-ending learning, we explicitly disallow the learning agentπfrom learning across multiple episodes or in multiple environments, which is closer to humanlearning. -> Is this just the same as saying you’re reset free and in a single environment? This sentence is a bit confusing.

One potential criticism of using simplified simulated worlds like JBW is why should we believe that insights that we get from JBW would carry over to the real world natural environments that NEL ideally cares about. Why is this actually representative of the real world? Because that is really what we care about with NEL. There is merit to simulation in this setting but only if we believe that either insights, algorithms or policies also hold in the real world and we can representatively model the worlds complexity. Can we verify this somehow?

Design
Does the user have control over all the agents? Or how are they programmed

I would move the details of procedural generation to the appendix, they’re a bit distracting from the point.

I would also tell the readers why things like scent, intensity, interaction etc are important early on, otherwise it’s confusing what their purpose is.

In general I quite like the setup, it seems like it has the sufficient amount of complexity in modality, interaction and multi-agent systems to be useful. I wonder if it’s also useful to introduce autonomous self-powered agents which move on their own in the environment and introduce dynamic non stationarities.

A little more description of the multi-agent, multi-task, curriculum stuff would be useful in the design section.

The reward functions are all sparse? Or do they need guidance to get to objects as well?

I’m still a bit confused about the interactions functions. Could those be described a bit further?

Perhaps a practical question is how does this relate to the work described in the BabyAI/Minigrid stuff from MILA and other simulated gridworld style environments with multiple agents and such.

Experiments:
In the case studies, are things multi-agent?
I wonder if in the reset-free experiment, if we just use dynamic agents in a multi-agent setup, would this just work?

Is it a little odd that the without occlusion performance comes back down to around the same as with occlusions?

Is the scent just perhaps misconfigured/too hard to learn from coz it never seems like it’s doing well with scent?

Overall, I like the paper and the introduced environment. I think it’s important to study scenarios such as the ones described here and this provides a tractable way to start. I am however concerned that the environments are too simplistics and perhaps too far from the real world for the insights to carry over to more realistic scenarios. Some suggestions would be to try and make the environment a bit more realistic and less toy so that insights might also more easily transfer to real world scenarios. But I think with some of the clarifications above and a bit more description, this would be a valuable contribution on topics which are not thought about enough in RL. I also think that actual visuals and videos on an actually accessible website would make it easier for the reviewers/readers to understand the importance of this. I'm currently listing it as a weak accept but I would like the authors to better clarify some of the points mentioned above, discuss how realistic the setting is and also provide us with videos of the environment to better gauge things.

**Experience Assessment:**

I have published one or two papers in this area.

**Review Assessment: Checking Correctness Of Derivations And Theory:**

N/A

**Review Assessment: Checking Correctness Of Experiments:**

I assessed the sensibility of the experiments.

**Review Assessment: Thoroughness In Paper Reading:**

I read the paper at least twice and used my best judgement in assessing the paper.

---

> ### Author Response · Authors · 2019-11-10
> **Response to Official Blind Review #3 Part 1/2**
>
> Thank you for the feedback and helpful comments! We address your concerns below, in the order in which they were listed.
>
> 1. Lifelong/Continual Learning vs Never-Ending Learning:
>
> Continual learning, lifelong learning, and never-ending learning (NEL) have considerable overlap and were oftentimes used interchangeably. The definition of NEL that we provide generalizes on the definition of lifelong learning provided by Chen and Liu 2018 (in “Lifelong Machine Learning, 2nd Edition”). However, since the two overlap to such a large degree, the JBW provides a good testbed for both never-ending and lifelong learning. Existing evaluation frameworks are lacking in many of the same ways for testing lifelong learning as they are for testing NEL. The never-ending, reset-free aspect of NEL is an important distinction with multi-task learning. To address this, we added a new paragraph to the introduction to address this.
>
> 2. “Is this just the same as saying you’re reset free and in a single environment? [The] sentence is a bit confusing.”:
>
> We revised the relevant sentence to “In never-ending learning, agents can only exist in a single environment that is reset-free (i.e., we explicitly disallow the agent π from learning across multiple episodes or in multiple environments, which is closer to human learning).”
>
> 3. “JBW is...too simplistic and perhaps too far from the real world…”:
>
> This criticism is common to many existing testbeds, and we agree that it is a valid one. We believe that simpler environments provide researchers with the ability to tackle problems in machine learning in a more controlled and isolated environment, without having to deal with the full generality and complexity of real-world environments. We believe there are lessons to be learned from working with these simpler environments that can be generalized to real-world environments. This was partly the reason why we defined a notion of complexity and why we advocate for testbeds of higher complexity, since they require learning algorithms with significantly better generalizability. For example, one potential avenue of research is to develop agents that maintain an internal model/representation of the world. A more task-independent and modality-independent model/representation would lead to more generalizable agent behavior. Developing these kinds of algorithms in the JBW could be very useful and applicable to other environments and real-world settings. In addition, relative to alternative testbeds, our proposed environment is a step closer to the real world in many respects, which we described in the paper. Due to space limitations, we did not add any sentences in our revisions to address this point, but if you recommend that we should, then we will do so.
>
> 4. “Does the user have control over all the agents? Or how are they programmed”:
>
> The JBW provides a programmatic interface for agents, providing functions to query the current vision and scent perception of each agent, as well as functions to direct agents to perform actions in the world. Thus, users write their own code for the agent’s decision logic by using these functions. If the user so chooses, it is possible to implement human control of agents via a keyboard-mouse interface, for example, on top of the provided programmatic interface. The JBW does not prevent the user from doing so. To make this more clear, we modified the “Interface” paragraph in Section 2.1 to read: “Users interact with the simulator programmatically. JBW provides functions to add or remove agents from the world, query the current vision and scent perception of each agent, and to direct agents to perform actions. Users can choose to add multiple agents to the world…”
>
> With respect to multi-user/multi-agent experiments, the JBW provides a simple access control mechanism which allows for privileges to be selectively granted/denied to users.
>
> 5. “I wonder if it’s also useful to introduce autonomous self-powered agents...”:
>
> This is a very interesting idea, and is definitely something users could do in the JBW. While we did not experiment with this ourselves, it’s a promising direction for future work, and is already supported in the JBW.
>
> 6. “A little more description of the multi-agent, multi-task, curriculum stuff would be useful...”:
>
> The design section is intended to focus on the Jelly Bean World and the simulator, rather than on the tasks and reward functions that can be defined within that world. Due to space constraints and the focus of the paper on the design of the JBW simulator and world generator, we provide a short discussion on the types of learning tasks that can be defined in the JBW in Section 3.

---

> ### Author Response · Authors · 2019-11-10
> **Response to Official Blind Review #3 Part 2/2**
>
> 7. “The reward functions are all sparse? Or do they need guidance to get to objects as well?”:
>
> The JBW allows users to define their own reward functions. We provide a handful of built-in reward functions presented in Section 3. The reward function that we refer to as “Collect” is indeed sparse. The other two functions (“Action” and “Explore”) are not sparse. Users are free to design their own reward schedules perhaps to help guide the learning of tasks with sparser rewards. We added the following sentence after this table to make this clearer: “We note that ‘Collect’ is a sparse reward function, whereas ‘Action’ and ‘Explore’ are not.”
>
> 8. “Could [interaction functions] be described a bit further?”:
>
> We added the following sentences after first defining the intensity and interaction functions in Section 2.1 (in the “Map” paragraph): “The intensity function characterizes the (log) probability of the existence of an item $I_i$ independent of other items in the world. The interaction function can be understood as a description of the (log) probability of the existence of an item $I_i$ given the existence of all other items $I_j$. For example, the interaction function can be used to increase the log probability of an item when it appears near other items, producing a clustering effect.”
>
> 9. BabyAI/Minigrid and Other Simulated Gridworlds:
>
> A big difference between the JBW and all other gridworld-style environments is that the JBW is designed to tackle NEL tasks that run “forever.” For example, all tasks in BabyAI seem to be finite and extending them to an infinite setting poses many of the challenges that the proposed JBW is designed to address. Furthermore, the JBW gives a much higher level of control over defining learning tasks and configuring the world (i.e., the distribution of items can be fully configured). It also supports non-stationary environments by design. We have updated the paper to also include an appendix section with a configuration that produces a non-stationary environment and a visualization thereof. These features together are unique to the JBW and not supported by any of the existing gridworld style environments you mentioned. We hope that this addresses your concern. We have also added a reference to BabyAI in Table 1 with a note on how it differs from the JBW.
>
> 10. Multi-Agent Experiments:
>
> The case studies do not feature multiple agents since we intended for them to highlight other aspects of the JBW, such as its complexity and the complementarity of vision and scent. But yes, the JBW is reset-free and supports multi-agent settings with dynamic agents. It is able to correctly handling the multiple simultaneous interactions via an efficient synchronization scheme, without hurting performance. We added the following sentence in the beginning of Section 4 to make this clearer: “Due to space constraints, the case studies focus on the single-agent setting.”
>
> 11. Performance with/without Occlusion:
>
> As shown in Figure 4, the performance of most RL agents eventually becomes really bad in the NEL setting. This is mainly due to the reset-free nature of the JBW and the fact that agents tend to get stuck in regions of the JBW that they have already exhausted. In this case, the “no-occlusion” agent is much better at collecting jelly beans early on but this means it more quickly collects all of the available jelly beans in a region of the map much faster and eventually starts getting stuck in that region being unable to find new jelly beans. This points out one of the important challenges for learning algorithms which the JBW aims to provide a testbed.
>
> 12. Is Scent too Hard:
>
> The heavy reliance on vision was due to the particular environment configuration that we used for the case studies. We agree that it is important to, at least, demonstrate the utility of scent as an orthogonal feature to vision. Since we were not able to fit these results in the main paper, we have added an appendix section with experimental results using a different environment configuration in which an agent learns to rely on scent as opposed to vision. In this section, we also added a discussion on how various differences in the environment could affect the relative utility of the vision and scent modalities. These results also demonstrate that it is possible to construct worlds in the JBW in which agents learn more effectively by using scent than by using vision. In this section, we also briefly discuss how certain design choice for the learning algorithm (such as using a neural architecture that supports some kind of memory) can affect the relative utility of the modalities.
>
> 13. Website with Visuals/Videos:
>
> Having a website to showcase the testbed is a great idea. While we did not implement one for the submission, we do want to do so before the conference. For now, we have added an additional figure in the new appendix section with the non-stationary environment with additional visualization of a generated environment.

---

### Official Review · AnonReviewer2 · 2019-10-23
**Official Blind Review #2**

**Rating:** 6

**Review:**

1. Summary

The authors introduce a simulator (JBW) with the goal of supporting continual learning. They demonstrate that RL agents struggle with a lot of the tasks in JBW. The majority of the paper describes the technical details of JBW, and show that RL agents can struggle to solve continually changing tasks in JBW.

2. Decision (accept or reject) with one or two key reasons for this choice.

I'm borderline. It is valuable to have environments that support continual learning, although the experimental investigation into different forms of non-stationarity would be more informative.

Re new implementations: the continual learning setting is certainly important and interesting, but existing environments (see BabyAI, https://arxiv.org/abs/1810.08272 (focus on NLP)) do feature multiple tasks and it is not hard to augment these to run `forever'.

**Experience Assessment:**

I have published one or two papers in this area.

**Review Assessment: Checking Correctness Of Derivations And Theory:**

N/A

**Review Assessment: Checking Correctness Of Experiments:**

I assessed the sensibility of the experiments.

**Review Assessment: Thoroughness In Paper Reading:**

N/A

---

> ### Author Response · Authors · 2019-11-10
> **Response to Official Blind Review #2**
>
> Thank you for the feedback and helpful comments!
>
> We agree that additional examples of non-stationarity would be valuable. To this end, we have added an appendix section with a configuration that produces a non-stationary environment and a visualization thereof. The key idea is that to induce non-stationarity, we can generate environments where the distribution of items is not stationary in space. This is done by using non-stationary intensity and interaction functions (refer to Eq. 1) for the example we added in the appendix. Note that non-stationarity can also be induced by multi-agent settings, which the JBW already supports.
>
> We also thank you for the BabyAI reference. We have included it in the related work discussion although we are not sure how it could be augmented for continual learning tasks that run “forever.” All tasks in BabyAI seem to be finite and extending them to an infinite setting poses many of the challenges that the proposed JBW is designed to address.

---

### Author Response · Authors · 2019-11-10
**Summary of Response to Reviews**

We thank all reviewers for the helpful feedback and comments!

We respond to each one individually, but we would also like to emphasize that in order to address some of the concerns that were shared across multiple reviewers, we added Sections A.6 and A.7 in our appendix, providing more details on the relative utility of scent and vision and on non-stationary environment configurations, respectively. We have also added clarifications in the introduction and across other sections of the paper addressing the concerns of our reviewers.

---

### Decision · Program_Chairs · 2019-12-19

**Decision:**

Accept (Poster)

**Comment:**

This paper proposes a flexible environment for studying never ending learning. During the discussion period, all reviewers found the paper to be borderline.

Pros:
- we don't have good lifelong or never-ending RL environments, and this paper seems to provide one
- includes a number of interesting features such as multiple input modalities, non-episodic interactions, flexible task definitions

Cons:
- procedurally generated, toy environment
- unclear if the environment reflects the characteristics of real world NEL problems

In the balance, I think the environments add value to the RL community, and being presented at ICLR would increase its visibility.